# Genomic Characterization of Potential Plant Growth-Promoting Features of *Sphingomonas* Strains Isolated from the International Space Station

Jonathan Lombardino,[a] Swati Bijlani,[b] Nitin K. Singh,[c] Jason M. Wood,[c] Richard Barker,[a] Simon Gilroy,[a] Clay C. C. Wang,[b] Kasthuri Venkateswaran[c]

[a]University of Wisconsin-Madison, Madison, Wisconsin, USA
[b]University of Southern California, Los Angeles, California, USA
[c]Biotechnology and Planetary Protection Group, Jet Propulsion Laboratory, California Institute of Technology, Pasadena, California, USA

Jonathan Lombardino and Swati Bijlani contributed equally to this article. Author order was determined alphabetically by first name.

**ABSTRACT** In an ongoing microbial tracking investigation of the International Space Station (ISS), several *Sphingomonas* strains were isolated. Based on the 16S rRNA gene sequence, phylogenetic analysis identified the ISS strains as *Sphingomonas sanguinis* ($n = 2$) and one strain isolated from the Kennedy Space Center cleanroom (used to assemble various Mars mission spacecraft components) as *Sphingomonas paucimobilis*. Metagenomic sequence analyses of different ISS locations identified 23 *Sphingomonas* species. An abundance of shotgun metagenomic reads were detected for *S. sanguinis* in the location from where the ISS strains were isolated. A complete metagenome-assembled genome was generated from the shotgun reads metagenome, and its comparison with the whole-genome sequences (WGS) of the ISS *S. sanguinis* isolates revealed that they were highly similar. In addition to the phylogeny, the WGS of these *Sphingomonas* strains were compared with the WGS of the type strains to elucidate genes that can potentially aid in plant growth promotion. Furthermore, the WGS comparison of these strains with the well-characterized *Sphingomonas* sp. LK11, an arid desert strain, identified several genes responsible for the production of phytohormones and for stress tolerance. Production of one of the phytohormones, indole-3-acetic acid, was further confirmed in the ISS strains using liquid chromatography-mass spectrometry. Pathways associated with phosphate uptake, metabolism, and solubilization in soil were conserved across all the *S. sanguinis* and *S. paucimobilis* strains tested. Furthermore, genes thought to promote plant resistance to abiotic stress, including heat/cold shock response, heavy metal resistance, and oxidative and osmotic stress resistance, appear to be present in these space-related *S. sanguinis* and *S. paucimobilis* strains. Characterizing these biotechnologically important microorganisms found on the ISS and harnessing their key features will aid in the development of self-sustainable long-term space missions in the future.

**IMPORTANCE** *Sphingomonas* is ubiquitous in nature, including the anthropogenically contaminated extreme environments. Members of the *Sphingomonas* genus have been identified as potential candidates for space biomining beyond earth. This study describes the isolation and identification of *Sphingomonas* members from the ISS, which are capable of producing the phytohormone indole-3-acetic acid. Microbial production of phytohormones will help future *in situ* studies, grow plants beyond low earth orbit, and establish self-sustainable life support systems. Beyond phytohormone production, stable genomic elements of abiotic stress resistance, heavy metal resistance, and oxidative and osmotic stress resistance were identified, rendering the ISS *Sphingomonas* isolate a strong candidate for biotechnology-related applications.

Address correspondence to Kasthuri Venkateswaran, kjvenkat@jpl.nasa.gov.

The authors declare no conflict of interest.

KEYWORDS *Sphingomonas*, International Space Station, phytohormones, plant growth promotion

*S*phingomonas species are found in a variety of habitats, including soil, marine sediments, wastewater, and freshwater, in addition to the habitats contaminated with dyes, herbicides, and other pollutants (1–3). They possess diverse catabolic capabilities; for instance, assimilation of different types of complex sugars, including polysaccharides, was reported (2). Some species are tolerant to copper, resulting in biofilm formation and eventually surface corrosion (4), and some aid in herbicide mineralization or degradation of pollutants (5–7). *Sphingomonas paucimobilis* has been isolated from the International Space Station (ISS) potable drinking water (8), ISS surfaces (9), and hospital clinical specimens and is known to cause rare infections in immunocompromised individuals (2, 10, 11).

Recent reports have identified the role of some *Sphingomonas* species in plant growth promotion (5, 12–14). Microorganisms aid in plant growth promotion via different mechanisms, including secreting metabolites like phytohormones, solubilizing phosphate, fixating nitrogen, inducing stress response pathways, or acting as biocontrol agents (5, 14–19). Phytohormones play a significant role in plant metabolism and defense mechanisms (20). For instance, indole-3-acetic acid (IAA) is responsible for promoting cell division, elongation, and differentiation (21) and is involved in a wide range of plant processes ranging from responses to light and gravity to heavy metal stress tolerance (22, 23). Recent studies have detailed the role of *Sphingomonas* species in plant growth promotion via the production of phytohormones and increased stress tolerance. *S. paucimobilis* ZJSH1 produces phytohormones like IAA, salicylic acid, zeatin, and abscisic acid and helps in nitrogen fixation (14). *Sphingomonas* sp. LK11, isolated from the leaves of the arid medicinal plant *Tephrosia apollinea*, was shown to produce auxins and gibberellins and was reported to increase stress tolerance to salinity and cadmium in tomato plants (12, 15, 24). In addition, genomic analysis of *Sphingomonas* sp. LK11 also revealed genes for hydrogen peroxide production that enhance plant growth, seed germination, and root colonization, genes for phosphate uptake, osmotic stress, and heavy metal stress (15). Genomic analysis of *Sphingomonas panacis* DCY99[T] isolated from the roots of *Panax ginseng* identified genes for IAA production, phosphate solubilization, heavy metal stress tolerance, and degradation of phenolic compounds (5). Furthermore, *Sphingomonas* sp. Cra20 was shown to improve plant growth under drought stress conditions by altering the rhizosphere microbial community (13). The ability of microorganisms to produce phytohormones thus offers the potential to manipulate plant growth and development but without the need to supplement hormones externally to the plants.

In an ongoing microbial tracking experiment on the ISS, several microbial strains belonging to *Sphingomonas sanguinis* (*n* = 2) were isolated from the ISS, and a third strain identified as *S. paucimobilis* was cultured from the Kennedy Space Center, Payload Hazardous Servicing Facility (KSC-PHSF). After the isolation of *Sphingomonas* strains from the ISS (25, 26), a complete genome (~4.4 Mbp) was also assembled from the shotgun reads retrieved from the same ISS environmental surface (26), resulting in the production of a metagenome-assembled genome (MAG) belonging to *S. sanguinis*. Given the persistence of *Sphingomonas* bacteria among spaceflight-relevant surfaces, and a growing body of evidence for *Sphingomonas* strains to promote plant growth, this study applies comparative genomic analyses to identify genetic determinants that relate to plant growth-promoting features (PGPF) and highlights those that may be adaptive to growth on spaceflight-relevant environments and surfaces. Our findings report the presence of several determinants for the production of plant phytohormones, including several gene products promoting resistance to sources of abiotic stress, with emphasis on oxidative/osmotic stress resistance, heavy metal tolerance, and adaptation to sources of radiation. Further, the production of a phytohormone, IAA, by these isolates was confirmed using metabolomics. Moreover, intersection analysis of functional annotations and orthologous

groups from the *Sphingomonas* species from spaceflight-relevant surfaces (KSC-PHSF, ISS) revealed potential differences in gene contents involved in the horizontal gene transfer that are notably absent in their respective type strains. Taken together, our results show how characterization of microorganisms with respect to their ability to promote plant growth, including vegetables that can be used as a food source, will contribute to the development of self-sustainable long-term space missions in the future.

## RESULTS

**Abundance of *Sphingomonas* species in the ISS.** Among the cultivable bacteria, three colonies obtained from ISS surfaces (IIF7SW-B3A and IIF7SW-B5) and one from KSC (FKI-L5-BR-P1) were identified as *Sphingomonas* species. All ISS strains were *S. sanguinis*, and the KSC strain was *S. paucimobilis*. The whole-genome sequencing (WGS) and the draft genomes assembled were further used to confirm the identity of the species based on average nucleotide identity (ANI; Table 1), which was calculated to be ~99% with the nearest type strain. Genomic characteristics and other details of the annotation are given in Table 1. The metagenomic data analysis identified 23 *Sphingomonas* species from flight 2, all of which were isolated from location 7, that is, a panel near the portable water dispenser (LAB) (Fig. 1). In addition, the reads obtained from flight 1 belonging to a single species, *Sphingomonas sanxanigenens*, were also seen in samples from flights 2 and 3 (Fig. 1). However, its location on the ISS varied across different flights. Reads obtained from flight 3 also showed the presence of *S. paucimobilis* from location 4, that is, the dining table (node 1). The metagenomic reads thus obtained from the ISS show that *Sphingomonas* species persisted across all three flights, with *S. sanguinis* being abundant among all *Sphingomonas* species in flight 2. Furthermore, the MAG sequence (ISS-IIF7SWP) exhibited >99.6% ANI with the type strain of *S. sanguinis* (Table 1). The MAG was derived from the exact location on the ISS where *S. sanguinis* isolates IIF7SW-B3A and IIF7SW-B5 were retrieved using cultivation techniques. It was also confirmed that the MAG genome had >99% ANI value with the genomes of the *S. sanguinis* strains isolated from the ISS. The use of the culture-dependent and -independent methods thus confirms the presence of *S. sanguinis* on the ISS.

**Plant growth-promoting features in *Sphingomonas* species isolated from spaceflight environments.** BLAST-based genome comparisons visualized using BLAST Ring Image Generator (BRIG) revealed relatively high levels of identity shared across each spaceflight genome and their type strains compared against the *Sphingomonas* sp. LK11 genome (Fig. 2). Additionally, patterns of GC content for each *S. sanguinis* (66.1 to 66.2%) and *S. paucimobilis* (65.5%) genome were highly similar to that of LK11 (66.09%). Subsequent pangenome comparisons conducted by Anvi'o revealed a substantial core genome and further support both shared nucleotide and functional identity among these spaceflight *Sphingomonas* genomes, their type strains, and the endophyte *Sphingomonas* sp. LK11 (Fig. 3).

Due to the high degree of functional identity shared between each of the space *Sphingomonas* species and *Sphingomonas* sp. LK11, functional annotations for each of the spaceflight *Sphingomonas* genomes and their type strains were conducted to identify putative PGPF. Functional analysis of these outputs revealed that many PGPFs proposed for *Sphingomonas* sp. LK11 (15) are likely also present in the analyzed *S. sanguinis* and *S. paucimobilis* genomes. Annotations reported from the EggNOGv5 database suggest that several gene products proposed to promote phosphate and sulfur assimilation in plants are also present in these select *Sphingomonas* strains, albeit with some species-specific patterns. Pathways associated with phosphate uptake metabolism and solubilization in soil appear to be conserved across all the selected *S. paucimobilis* and *S. sanguinis* strains; however, gene products associated with sulfate/thiosulfate and hydrogen production may be exclusive to the selected *S. sanguinis* strains. Furthermore, several pathways proposed to promote plant resistance to abiotic stress, including the heat/cold shock response, heavy metal resistance, and oxidative and osmotic stress resistance, appear to be shared between these selected *S. sanguinis* and *S. paucimobilis* genomes.

Previous reports suggested that responses to oxidative stress in *Sphingomonas* sp.

**TABLE 1** Genomic characteristics of *Sphingomonas* strains isolated from the ISS and cleanroom

| Strain | NCBI genome accession no. | Isolation source | Genome size (Mb) | ANI value (%)[b] | No. of: | | | | | | |
|---|---|---|---|---|---|---|---|---|---|---|---|
| | | | | | Total genes[c] | Total protein coding genes[c] | RNA genes[c] | rRNA | tRNA | ncRNA | Pseudogenes[c] |
| *Sphingomonas sanguinis* IIF7SW-B3A | JABEOW000000000 | ISS, lab 3 overhead | 3.9 | 99.77 | 3,768 (100%) | 3,659 (97.11%) | 58 (1.54%) | 3 | 52 | 3 | 51 (1.35%) |
| *Sphingomonas sanguinis* IIF7SW-B5 | JABEOV000000000 | ISS, lab 3 overhead | 4.3 | 99.68 | 4,178 (100%) | 4,047 (96.86%) | 58 (1.39%) | 3 | 52 | 3 | 73 (1.75%) |
| *Sphingomonas paucimobilis* FKI-L5-BR-P1 | JABEOU000000000 | KSC-PHSF cleanroom floor[a] | 4.5 | 99.42 | 4,357 (100%) | 4,182 (95.98%) | 54 (1.24%) | 3 | 48 | 3 | 121 (2.78%) |
| *Sphingomonas sanguinis* ISS-IIF7SWP (MAG) | JABYQV000000000 | ISS, lab 3 overhead | 4.4 | 99.64 | 4,261 (100%) | 4,149 (97.37%) | 57 (1.34%) | 3 | 51 | 3 | 55 (1.29%) |

[a]KSC-PHSF cleanroom floor: Kennedy Space Centre- Payload Hazardous Servicing Facility.
[b]ANI values shown are calculated against their respective type strains *S. sanguinis* NBRC 13937 and *S. paucimobilis* NCTC 11030.
[c]% Values in parentheses indicate the proportion of genes among the total number of genes.

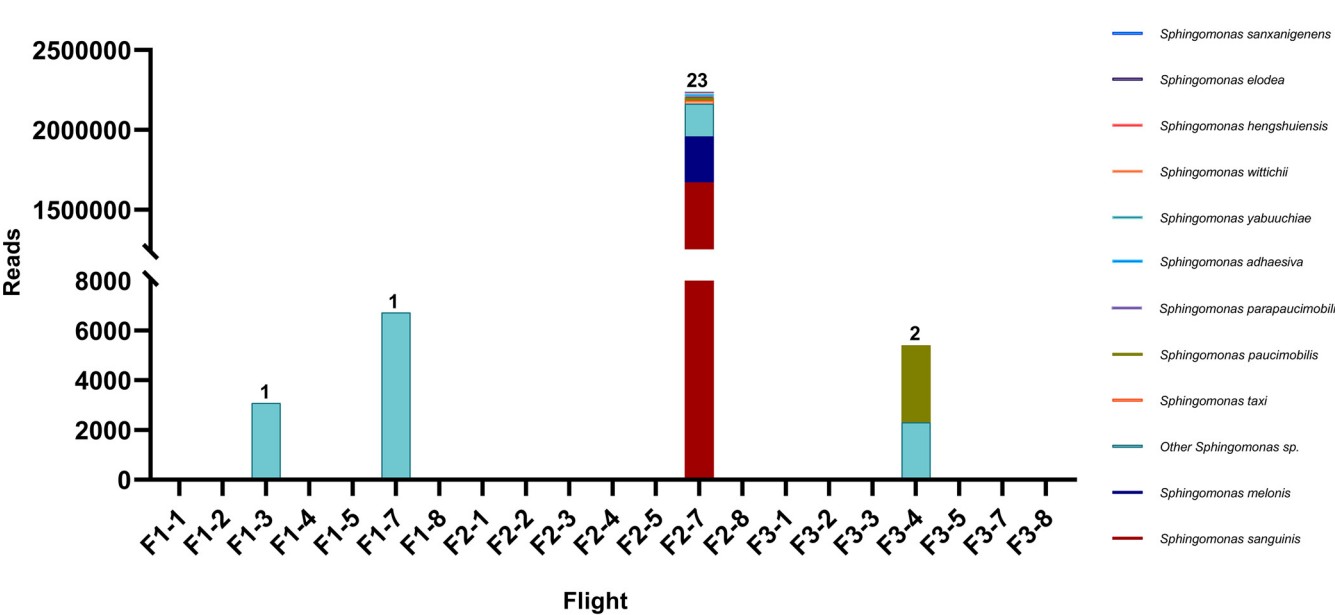

**FIG 1** The abundance of *Sphingomonas* species across different flights and locations on the ISS based on metagenome-derived genomic reads. The numbers on top of bars represent the number of *Sphingomonas* species identified from that particular location in the corresponding flight number (F1, F2, or F3). Different locations on the ISS from where samples were collected: location 1, port panel next to cupola (node 3); location 2, waste and hygiene compartment (node 3); location 3, advanced resistive exercise device (ARED) foot platform (node 3); location 4, dining table (node 1); location 5, zero G stowage rack (node 1); location 7, panel near portable water dispenser (LAB); and location 8, port crew quarters, bump out exterior aft wall (node 2).

LK11 may promote plant host tolerance to abiotic stress via the amelioration of reactive oxygen species (ROS) (12, 15, 27). Therefore, specific gene products related to oxidative stress resistance in space-associated *Sphingomonas* strains were further identified. Analysis of annotations generated by PGAP and eggNOG-MapperV2 suggested that space-associated *Sphingomonas* strains harbor multiple copies of several classes of antioxidant enzymes, including superoxide dismutases, catalases, peroxidases, glutathione S-transferases, and glutaredoxins. Next, we sought to identify metabolic pathways suggested to reduce the abundance of ROS, for instance, carotenoids, which are known for their antioxidant activity (28). Examination of PGAP and eggNOG annotations suggest that all *Sphingomonas* strains analyzed in this study possess the genetic repertoire (*crtB*, *crtI*, *crtY*, *crtZ*) to produce the carotenoid zeaxanthin from the precursor geranylgeranyl pyrophosphate (GGPP) (Fig. 4A). Further analysis also revealed the presence of an unnamed sterol desaturase family protein, often proximate to these gene clusters, which shared high identity to the *crtG* gene product identified in *Sphingomonas elodea* ATCC 31461 (74% to 75% identity) (29). *CrtG* converts zeaxanthin to produce the yellow pigment nostoxanthin, a common carotenoid produced by *Sphingomonas* (30). Likewise, two pathways for the production of the nonreducing disaccharide trehalose were also identified in both the *S. sanguinis* and *S. paucimobilis* strains. Trehalose serves a variety of functions across organisms, acting as a carbon source, a compatible solute, and a metabolite that provides protection against a variety of abiotic stressors (31). These results suggest that *Sphingomonas* strains can potentially synthesize trehalose from the *otsA/B* pathway, wherein a two-step mechanism catalyzes the formation of trehalose 6-phosphate from the condensation of glucose 6-phosphate and UDP-glucose residues (*otsA*) and subsequent dephosphorylation of trehalose 6-phosphate into the disaccharide trehalose (*otsB*). Additionally, trehalose could also be produced in these bacteria through catabolism of $\alpha$(1,4) glucose polymers into trehalose via the *treY/Z* pathway (Fig. 4B).

These findings suggest that the tested *Sphingomonas* species, including the strains sequenced from spaceflight-relevant surfaces, contain several pathways likely to confer resistance to a variety of abiotic stressors, especially with respect to several antioxidant

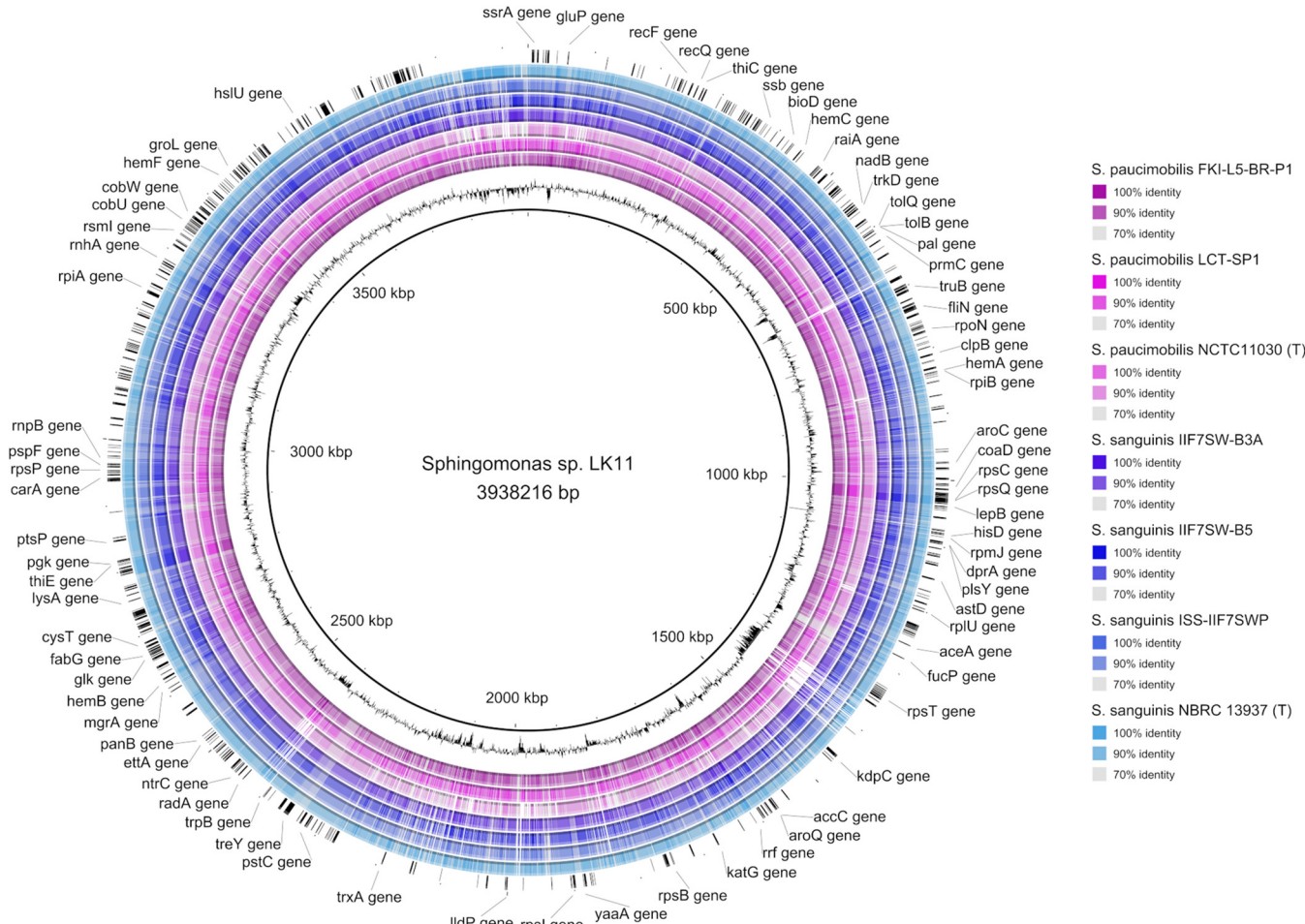

**FIG 2** Circular genome representations were constructed using BLAST Ring Image Generator (BRIG) version 0.95, with *Sphingomonas* sp. LK11 selected as the reference genome (chromosome and plasmids). The innermost ring (black) shows the GC content of the reference *Sphingomonas* sp. LK11 genome. Subsequent rings illustrate the percent identity of the selected *Sphingomonas* genomes calculated using blastn (E value: 10, lower identity cutoff: 70%, upper identity cutoff: 90%). Starting from ring two, the order of the illustrated genomes is as follows: 2, *S. paucimobilis* FKI-L5-BR-P1; 3, *S. paucimobilis* LCT-SP1; 4, *S. paucimobilis* NCTC 11030[T]; 5, *S. sanguinis* IIF7SW-B3A; 6, *S. sanguinis* IIF7SW-B5; 7, *S. sanguinis* ISS-IIF7SWP (MAG); 8, *S. sanguinis* NBRC 13937[T]. The outermost ring displays randomly selected annotations originating from the *Sphingomonas* sp. LK11 chromosome.

enzymes and metabolites. The presence of these features supports the potential role of these newly sequenced *Sphingomonas* strains to promote plant growth and increase plant stress tolerance.

**Indole-3-acetic acid production by *S. sanguinis* and *S. paucimobilis* strains isolated from the ISS.** One of the sources of IAA biosynthesis is the amino acid tryptophan, and analysis of the orthologous groups identified by the eggNOGv5 database suggested that the selected *Sphingomonas* strains possess the required tryptophan biosynthesis genes (*trpA,B,C,D,E,F*), consistent with observations from previous reports in the literature (5, 15). Leveraging the inclusion of several KEGG orthologs, enzyme classes, and KEGG reactions outputs via eggNOG-MapperV2, potential pathways for these *Sphingomonas* to synthesize IAA from tryptophan are proposed (Fig. 4C). Analysis of the functional annotations transferred from the EggNOGv5 database pointed to a group of carbon-nitrogen hydrolase family proteins that may act as putative nitrilases (E.C 3.5.5.1), converting indole-3-acetonitrile directly to IAA. Similarly, each of the analyzed *Sphingomonas* strains harbors a flavin adenine dinucleotide (FAD)-dependent oxidoreductase (putative monoamine oxidase EC 1.4.3.4), which likely promotes the conversion of tryptamine into indole-3-acetaldehyde. However, subsequent steps allowing the conversion of indole-3-acetaldehyde to IAA via an unnamed aldehyde dehydrogenase family protein (EC 1.2.1.3) were exclusive to the

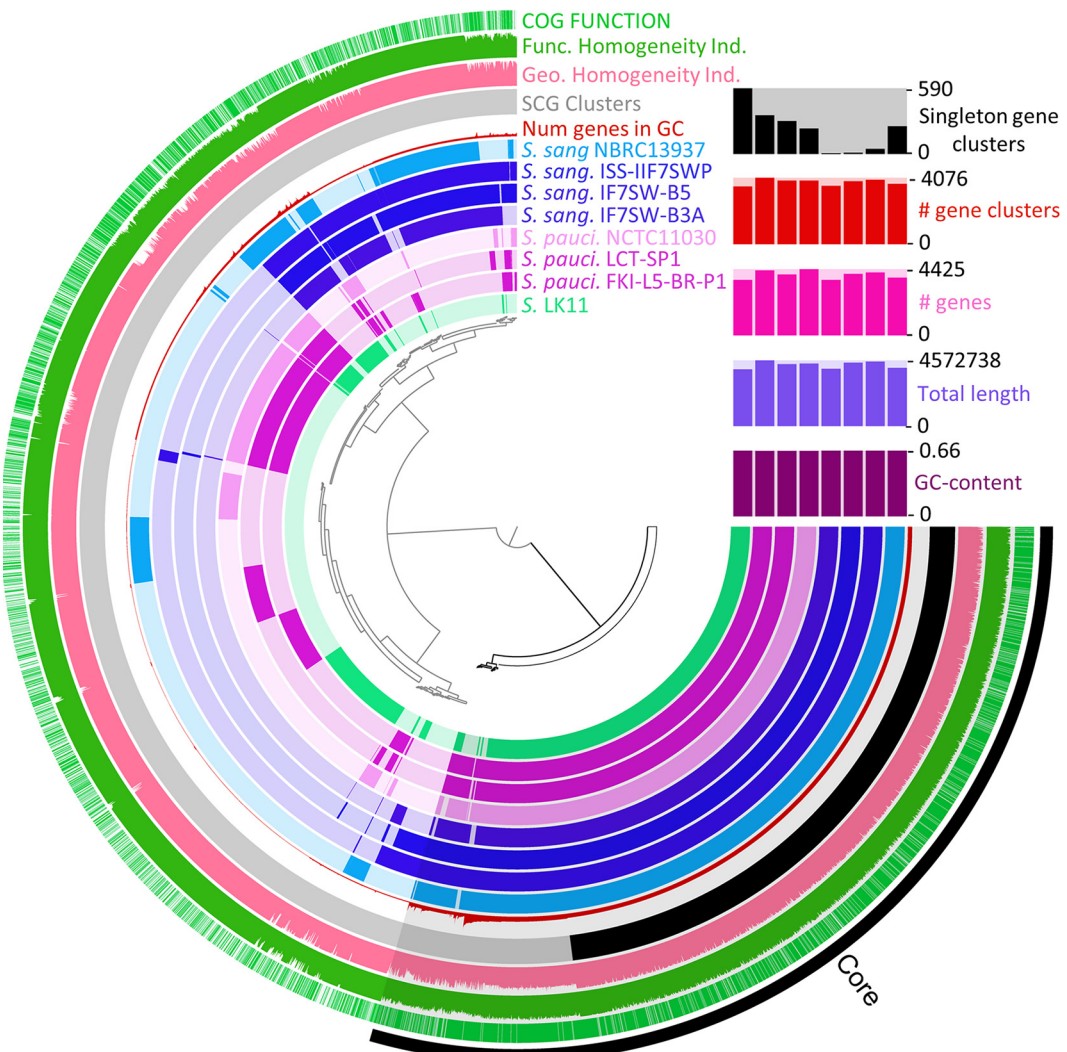

**FIG 3** Pangenome analysis of *S. sanguinis*, *S. paucimobilis* genomes obtained from space-relevant surfaces with their type strains, and *Sphingomonas* sp. LK11, generated by the Anvi'o software. Primary functional annotation in Anvi'o was conducted using the anvi-script-FASTA-to-contigs-db and anvi-run-ncbi-cogs commands. Subsequent pangenome gene clustering was carried out using blastp via the anvi-pan-genome command (–num-threads 2, –mcl-inflation 6, –min-bit 0.5, –use-ncbi-blast). Ordering of the pangenome display was determined using a Euclidean distance clustering algorithm and the provided ward linkage method. Beginning from the innermost ring and moving outward, rings 1 to 8 correspond to gene clusters identified in each of the analyzed *Sphingomonas* genomes in the following order: 1, *Sphingomonas* sp. LK11; 2, *S. paucimobilis* FKI-L5-BR-P1; 3, *Sphingomonas* LCT-SP1; 4, *S. paucimobilis* NCTC 11030[T]; 5, *S. sanguinis* IIF7SW-B3A; 6, *S. sanguinis* IIF7SW-B5; 7, *S. sanguinis* ISS-IIF7SWP (MAG); 8, *S. sanguinis* NBRC 13937[T]. Ring 9 shows a density plot of the number of genes within the identified gene clusters, and ring 10 highlights single-copy gene clusters (black). Rings 11 and 12 correspond to the geometric homogeneity index and functional homogeneity indices of similarity, with the former accounting for gaps in alignments, while the latter scores functional similarities by residue within the aligned sequences. The outermost ring represents the location of known Clusters of Orthologous Genes (COG) functional categories. The right-most bar chart corresponds to the number of singleton gene clusters, number of gene clusters, number of genes, total length, and GC content for each genome present in the analysis.

analyzed *S. paucimobilis* genomes. Further, the presence of indole-pyruvate ferredoxin, an enzyme which is responsible for the conversion of indole-3-pyruvate into S-2-(indol-3-yl)acetyl coenzyme A (acetyl-CoA), was also exclusive to *S. paucimobilis*; however, it is unclear as to how *S. paucimobilis* may further alter S-2-(indol-3-yl)acetyl-CoA into IAA. While two putative mechanisms across both *S. sanguinis* and *S. paucimobilis* that may directly yield IAA from tryptophan-derived metabolites are supported by the results of functional annotation, we failed to identify canonical mechanisms for the formation of the upstream reactants in tryptophan metabolism, suggesting that novel mechanisms for IAA formation may occur in *Sphingomonas* and species-dependent

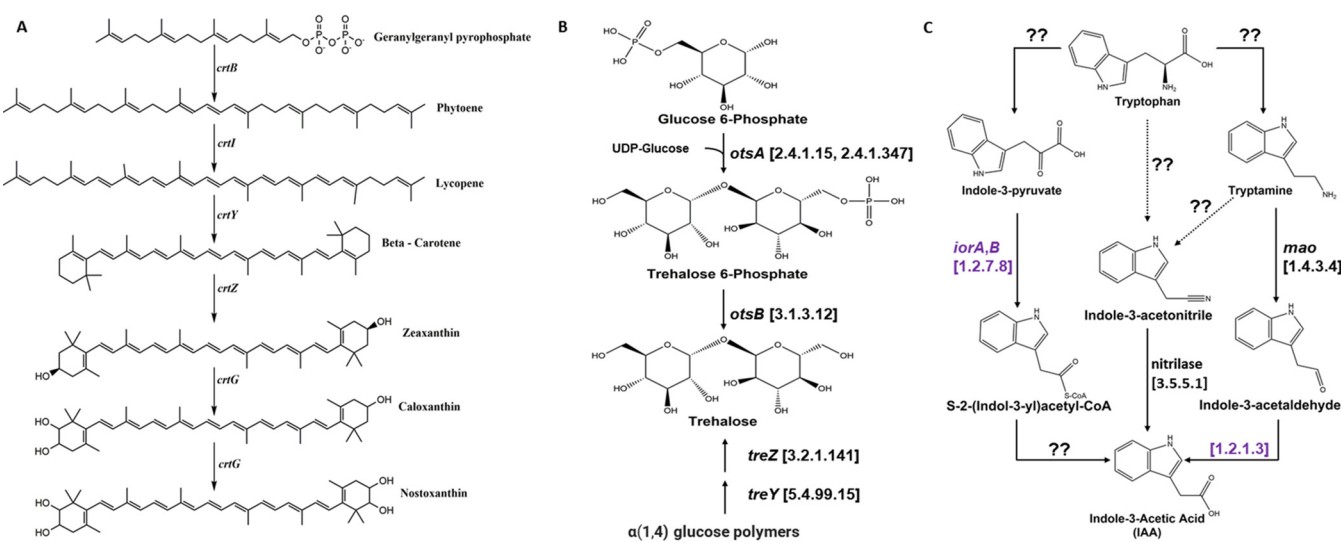

**FIG 4** (A) Zeaxanthin and nostoxanthin pathway. (B) Genes discussed relevant to trehalose metabolism in *S. sanguinis* and *S. paucimobilis*. In the first step of the *otsA/B* pathway, a condensation reaction between glucose 6-phosphate and UDP-glucose synthesizes trehalose 6-phosphate (*otsA*). Second, a dephosphorylation of trehalose 6-phosphate catalyzed by *otsB* yields the product trehalose. Additionally, trehalose can also be formed from the rearrangement of the α(1,4) glycosidic bonds of glucose polymers to α(1) glycosidic bonds via *treY*, allowing for the hydrolysis of the terminal trehalose disaccharide via *treZ*. (C) Proposed pathway for IAA production in *S. paucimobilis* and *S. sanguinis* using functional predictions generated by eggNOG-MapperV2. Enzyme classes are denoted in brackets, with gene, enzyme, or protein names provided where appropriate. Purple coloring denotes identified enzyme classes that were identified only in the analyzed *S. paucimobilis* spaceflight strains. Question marks (??) denote yet-to-be-validated reaction mechanisms in the proposed pathway. Chemical structures were constructed using the ChemDraw software.

pathways may supplement a shared metabolic pathway. Interestingly, while an auxin export carrier family protein was identified in each of the analyzed *S. paucimobilis* genomes, no such protein was identified in *S. sanguinis*.

Since the genomic analysis of the spaceflight isolates revealed genetic signatures for the production of several plant growth-promoting hormones, metabolomic characterization was carried out to identify the phytohormone IAA. All *Sphingomonas* isolates obtained from spaceflight-relevant surfaces were cultured in the presence of tryptophan, and metabolites extracted from the culture filtrate were analyzed using high-performance liquid chromatography-diode array detection-mass spectrometry (HPLC-DAD-MS). The MS data analysis showed the presence of IAA among the metabolites extracted from all three strains, *S. sanguinis* IIF7SW-B3A, *S. sanguinis* IIF7SW-B5, and *S. paucimobilis* FKI-L5-BR-P1 (Fig. 5A). The production of IAA (molecular weight, M = 175.18 g/mol) by these isolates was confirmed by comparing the electrospray ionization mass spectrometry (ESI-MS) profile (positive mode: M+H = ~176 g/mol) of IAA standard with culture extracts, either alone or mixed with IAA standard (Fig. 5A). The ESI-MS profile of all the three culture extracts was found to be comparable with the standard. Further, the production of IAA was validated by tandem mass spectrometry to generate a mass fragmentation profile in positive mode, corresponding to M+H of ~176 g/mol. The three culture extracts resulted in a fragmentation pattern that was similar to the IAA standard, generating three different fragments, including a parent ion of ~176 g/mol (Fig. 5B). In contrast, the presence of IAA was not observed in nutrient media without tryptophan (data not shown). Liquid chromatography-mass spectrometry (LC-MS) data also revealed the presence of few other metabolites with molecular weights similar to those of other phytohormones; however, their positive identification will require further analysis, which is outside the scope of this study.

**Functional differences between ISS *Sphingomonas* isolates and their type strains.** Analysis of plant growth-promoting features suggested the potential for conserved functional characteristics between the spaceflight and type strains of *S. paucimobilis* and *S. sanguinis* genomes but with some distinctions. Therefore, functional differences were further explored through hierarchical clustering of the relative abundance of functional categories assigned by the eggNOG database (Fig. 6). Intriguingly, the selected

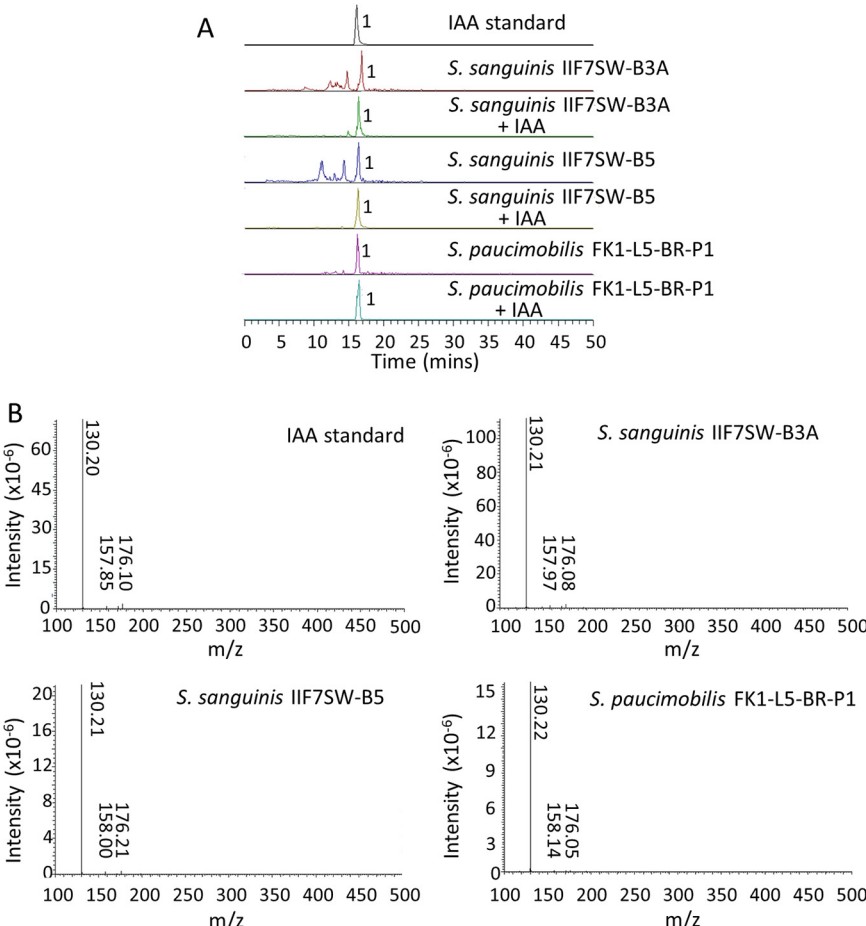

**FIG 5** (A) LC-MS profile of IAA produced by *S. sanguinis* and *S. paucimobilis* flight isolates, compared to the standard. (B) LC-MS/MS spectrum of IAA standard compared to that produced by *S. sanguinis* and *S. paucimobilis* flight isolates, all generating similar fragmentation patterns.

*Sphingomonas* genomes clustered primarily according to the environment in which they were sampled (terrestrial type strains versus spaceflight-relevant surfaces at KSC and the ISS) and then clustered according to their species assignment. Examining the relative abundance of each of the assigned functional categories, it is clear that a sizeable portion (~20% to 21%) of each of these selected *Sphingomonas* genomes lack known functional roles, highlighting novel gene products that are ripe for future bioinformatics and molecular characterization. Yet, differences in functional products corresponding to the category L, replication, recombination, and repair, are noticeably higher in genomes of the strains isolated from spaceflight-relevant surfaces.

Protein-level differences between the *Sphingomonas* strains obtained from spaceflight-relevant surfaces and their respective type strains are shown in Fig. 7A and Table S1. Intersection analysis was performed using UpsetR for the presence and absence of named protein products across each genome (Fig. 7B). Here, 1,488 named proteins were shared among all of the selected spaceflight-relevant *Sphingomonas* genomes and their respective type strains. In contrast, 71 named orthologous proteins were unique to the *S. paucimobilis* strains isolated from spaceflight-relevant surfaces, and an analogous 53 named orthologous proteins were found to be unique to *S. sanguinis* spaceflight strains. However, a sizable portion of the proteomes for each of these genomes lacked predicted protein names to use as unique terms for intersection analysis. Thus, a custom pileup was performed that grouped entries into unique keys consisting of common predicted taxa, predicted protein names, the eggNOG free text

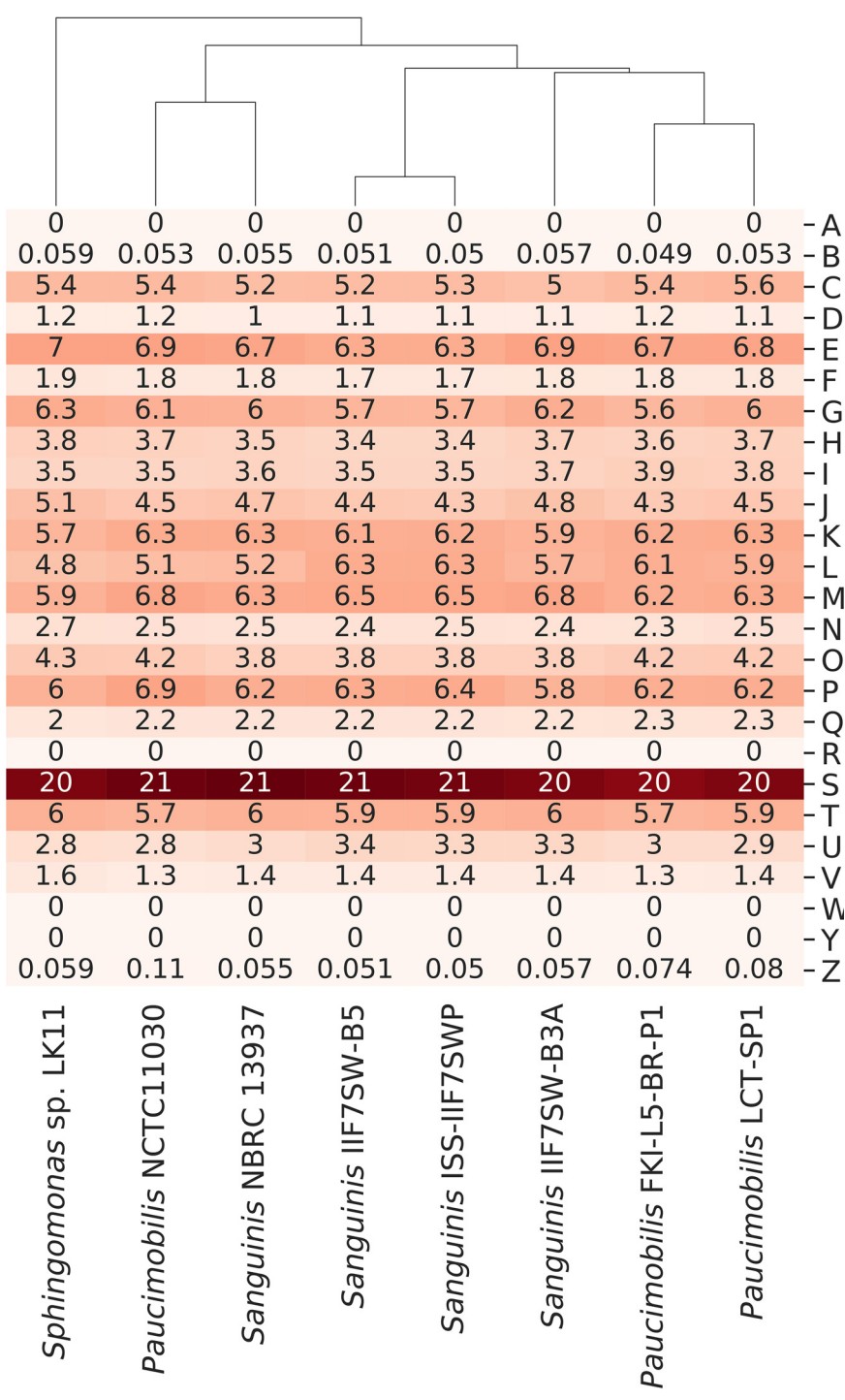

**FIG 6** Hierarchical clustering of the relative abundances for eggNOG functional categories for each of the selected *Sphingomonas* strains. Counts of each functional category were normalized to the total number of annotated proteins within each genome. The y-axis displays eggNOG functional category letter codes corresponding to the following: A, RNA processing and modification; B, chromatin structure and dynamics; C, energy production and conversion; D, cell cycle control, cell division, chromosome partitioning; E, amino acid transport and metabolism; F, nucleotide transport and metabolism; G, carbohydrate transport and metabolism; H, coenzyme transport and metabolism; I, lipid transport and metabolism; K, transcription; L, replication, recombination and repair; M, cell wall/membrane/envelope biogenesis; N, cell motility; O, posttranslational modification, protein turnover, chaperones; P, inorganic ion transport and metabolism; Q, secondary metabolites biosynthesis, transport and catabolism; R, general function prediction only; S, function unknown; T, Signal transduction mechanisms; U, intracellular trafficking, secretion, and vesicular transport; V, defense mechanisms; W, extracellular structures; Y, nuclear structure; Z, cytoskeleton.

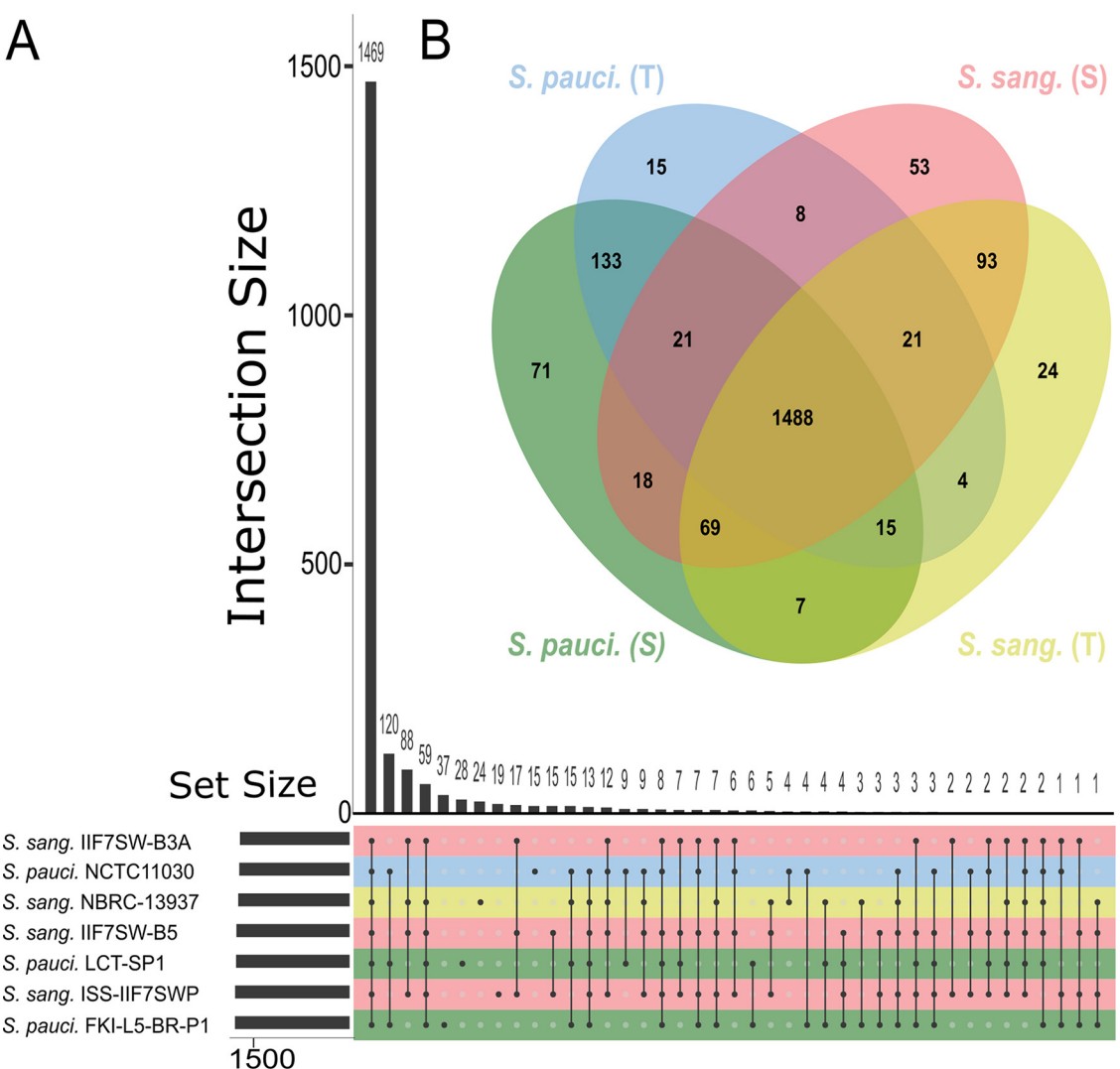

**FIG 7** (A) UpSetR plot showing the intersection of named eggNOG predicted proteins from *S. sanguinis* and *S. paucimobilis* genomes obtained from space-relevant surfaces and their respective type strains. Total named eggNOG protein counts from each strain are shown in a horizontal bar plot (left) and the number of shared named eggNOG proteins among the strains are represented in the vertical bar plot (right). (B) JVENN plot displaying the intersection of named EggNOG proteins according to their species and environment (S: space-relevant surface, T: type strain), where the named eggNOG proteins were pooled by category for simpler comparisons. Color schemes denote both the species and their environment of origin (green: *S. paucimobilis* space-relevant surface isolate, blue: *S. paucimobilis* type strain, pink: *S. sanguinis* space-relevant surface isolate, yellow: *S. sanguinis* type strain).

description, and their predicted functional category where unnamed proteins could be compared across multiple genomes. While this method enhanced comparisons of unnamed protein groups identified by eggNOGv5, precaution should be taken when comparing the results, as different seed orthologs appear to transfer slightly different descriptions; however, such an analysis greatly improved functional insights that would not have been captured via analysis of KEGG Orthology (KO) or named proteins alone. Analysis of the KO assigned by eggNOGv5 revealed that gene clusters associated with rhamnose metabolism, type IV secretion (*trb* operon), conjugal transfer pilus assembly proteins (*tra* operon), and heavy metal resistance genes are present in each of the spaceflight *S. sanguinis* strains and are notably absent from the type strain *S. sanguinis* NBRC 13937[T]. While *S. sanguinis* NBRC 13937[T] is not a complete genome, the ability to capture the presence or components of large biosynthetic clusters or operons is conducive to the assemblies' $N_{50}$ of 61,922 bp across 134 contigs of total length

4,053,092 bp. Similar patterns of differences in type IV secretion apparatus, conjugal transfer proteins, and rhamnose metabolism were also observed for the spaceflight-relevant surface strains from *S. paucimobilis* compared to those of their type strains, albeit to a lesser extent than the patterns reported for *S. sanguinis*. Further unique gene products identified in *S. paucimobilis* include a suite of multidrug efflux proteins, DNA repair proteins (*dinJ* and *sbcD*), catechol metabolism (*catABC*), genes involved in quorum sensing via homoserine lactone synthesis and export, and the natural transformation protein *comEA*. Both *S. sanguinis* and *S. paucimobilis* spaceflight isolates also showed the presence of genes that are involved in the biosynthesis of industrially important metabolites, including antibiotics (erythromycin and vancomycin) and anticancer compounds (betalains). Overall, our results point to functional differences in these spaceflight-relevant *S. paucimobilis* and *S. sanguinis* genomes compared to their respective type strains, many of which may provide adaptive advantages for survival during spaceflight or on spaceflight-relevant surfaces.

## DISCUSSION

This study shows the identification of putative PGPFs in *Sphingomonas* strains isolated from the ISS or in a cleanroom where spacecraft are assembled. The features involved in the plant growth promotion were identified based on comparative genomic analysis with a well-characterized endophyte *Sphingomonas* sp. LK11. Such comparative genomic analysis of the *S. sanguinis* IIF7SW-B3A, *S. sanguinis* IIF7SW-B5, *S. sanguinis* ISS-IIF7SWP, and *S. paucimobilis* FKI-L5-BR-P1 spaceflight genomes, along with their respective type strains, showed substantial genomic similarity to *Sphingomonas* sp. LK11, evidenced by high nucleotide identity and a large core genome shared among these bacteria. Leveraging extensive annotation provided by the eggNOG-MapperV2 software, PGPFs described previously for *Sphingomonas* sp. LK11 (14) were largely consistent with the results of comparative analysis conducted in this study. While minor species-specific differences in sulfur assimilation and hydrogen sulfide production were identified in *S. sanguinis*, all genomes shared similar putative PGPFs. Specifically, genomic analysis highlighted the presence of several enzyme classes that participate in ROS metabolism which appear to be present in multiple copies across each of the selected *Sphingomonas* genomes.

In addition, analysis of metabolic pathways of spaceflight *Sphingomonas* strains also suggested that each strain is likely capable of producing the carotenoids zeaxanthin and nostoxanthin. Zeaxanthin has been suggested to play a number of roles in both bacteria and plants, including photoprotection, detoxification of ROS, and resistance to ionizing radiation (32). Indeed, high levels of zeaxanthin production in the bacterium *Paracoccus zeaxanthinifaciens* were shown to promote resistance to ionizing, UV radiation, and hydrogen peroxide; however, resistance among the select *Sphingomonas* strains was weaker than that among *P. zeaxanthinifaciens* (33). Additionally, zeaxanthin also serves as a metabolic precursor for the formation of the plant phytohormone abscisic acid, prompting speculation that zeaxanthin-producing, plant-associated microbes may promote plant growth by supplying plant hosts with carotenoid derivatives such as zeaxanthin (34).

Further evidence for producing plant-relevant metabolites that confer antioxidant activity was also supported by the identification of the *otsA/B* and *treY/Z* pathways in each of the spaceflight *Sphingomonas* strains, strongly suggesting that these bacteria produce the nonreducing disaccharide trehalose. These results are in agreement with the identification of the *otsA/B* and *treY/Z* pathways in *Sphingomonas* sp. LK11 (15), as well as with previous work demonstrating that soybean plants inoculated with *Sphingomonas* sp. LK11, with and without exogenous trehalose, improved plant growth and stress tolerance under drought stress conditions induced via polyethylene glycol treatment (27). Likewise, transgenic introduction of the *Escherichia coli otsA/B* pathway in rice plant recipients yielded enhanced plant resistance to abiotic stress (35). In addition to promoting abiotic stress resistance in plants, trehalose production likely serves as an antioxidant against oxidative stress in microbes, conferring

resistance to hydrogen peroxide, and oxidative damage to proteins and lipids induced by ROS (36, 37). Due to its role as both an antioxidant and signaling molecule, trehalose production also elicits protective effects against sources of ionizing radiation (38–40).

Overall, these results strongly suggest that gene products identified across these spaceflight *Sphingomonas* strains support a model proposed for *Sphingomonas* sp. LK11 (a strain isolated from a desert plant), wherein excessive ROS produced by plant hosts is ameliorated through oxidative stress response pathways deployed by *Sphingomonas* strains, thereby increasing plant stress tolerance to abiotic environments (15). In addition, many of these ROS-metabolizing gene products discussed in the context of plant abiotic stress also contribute to increased resistance against ionizing radiation. Given evidence of oxidative stress responses deployed by microbial systems during spaceflight (41), and increased exposure to sources of ionizing radiation in extraterrestrial environments, we speculate that many of the discussed PGPFs likely present in spaceflight *S. sanguinis* and *S. paucimobilis* strains may be advantageous to their survival in the spaceflight environment.

Alongside several putative PGPFs, the genomic analysis suggested that IAA may be produced through three separate pathways in these organisms, albeit with some species-specific reactions. While all the space *S. sanguinis* and *S. paucimobilis* strains shared annotations for the conversion of indole-3-acetonitrile into IAA via a putative nitrilase enzyme, the production of IAA from the oxidation of indole-3-acetaldehyde via an unnamed aldehyde dehydrogenase family protein was absent in the spaceflight *S. sanguinis* strains. Yet, all *S. sanguinis* and *S. paucimobilis* strains may also produce indole-3-acetaldehyde from tryptamine, catalyzed by a monoamine oxidase enzyme. While functional analyses have proposed putative downstream reaction mechanisms to produce IAA, many of the upstream reaction pathways lack a well-characterized mechanism of action via known gene products or enzyme classes. Consistent with this study, not all genes involved in IAA biosynthesis were found in the genomes of *S. panacis* DCY99$^T$ and *Sphingomonas* sp. LK11 (5, 15), indicating that a novel pathway for IAA production in *Sphingomonas* strains is yet to be elucidated. Subsequent LC-MS analysis confirmed that each of the space *S. sanguinis* and *S. paucimobilis* isolates produced IAA, however, only when their growth medium was supplemented with tryptophan. The results obtained agree with other studies that have reported the production of IAA in *Sphingomonas* strains but, again, only in the presence of tryptophan in growth media (5, 12). In addition to the role of zeaxanthin and other plant metabolites in providing tolerance to certain stress conditions, IAA has also been reported to regulate several stress-related genes in *Escherichia coli* via accumulation of DnaK chaperone and trehalose (42). Some other studies have also reported an increase in trehalose accumulation upon exposure to IAA and found those cells to be more resistant to cold shock (43, 44). The chaperone DnaK has also been shown to protect *E. coli* from oxidative damage by preventing protein aggregation (45). This observation indicates that the production of IAA by space *Sphingomonas* strains might indirectly contribute to stress tolerance in the associated plants via similar mechanisms. Future applications of metabolomic and transcriptomic analyses under space or simulated microgravity conditions will likely be prudent for identifying patterns of metabolic flux in response to tryptophan supplementation, thereby elucidating potential mechanisms for IAA production in these bacteria.

Prompted by differences in functional compositions between *Sphingomonas* strains obtained from spaceflight-relevant surfaces (KSC, ISS, Shengzhou Rocket X), intersectional analysis of orthologous groups of spaceflight-relevant genomes against their type strains was performed. Analysis of KEGG-associated annotations provided by eggNOGv5 showed the presence of several genes in space-related *Sphingomonas* isolates that were not otherwise found in their respective type strains. *S. sanguinis* spaceflight isolates showed the presence of *tra* genes involved in pilus assembly and conjugal transfer (*traF*, *trbC*, *traU*, *traW*, *trbI*, *traC*, *traV*, *traB*, *traK*, *traE*, *traL*, and *traH*) and those involved in mating pair stabilization (*traG* and *traN*), which were absent in the type strain (46). Some groups have previously reported the occurrence of conjugal transfer in bacteria under microgravity conditions (47–49). Therefore, the presence of

*tra* genes in *S. sanguinis* spaceflight isolates indicates the existence of conjugal transfer under space conditions that might also be responsible for acquiring genetic materials from related bacterial species.

In addition, *S. paucimobilis* FKI-L5-BR-P1 showed the presence of genes involved in quorum sensing: *lasI* (acyl-homoserine lactone synthase), acyl-homoserine lactones (AHLs) biosynthesis, and LuxR family transcriptional regulator. AHLs act as signaling molecules through which LuxR regulates the expression of a number of genes, some of which are potentially associated with virulence (50). AntiSMASH analysis of *S. paucimobilis* FKI-L5-BR-P1 spaceflight isolate also showed the presence of a gene cluster (region 6.1; Fig. S1) involved in the synthesis of a homoserine lactone (Hserlactone).

Further, KEGG analysis also showed the presence of some genes involved in the production of industrially important metabolites in the spaceflight isolates. For instance, both *S. sanguinis* and *S. paucimobilis* spaceflight isolates showed the presence of modules associated with the biosynthesis of antibiotics like erythromycin A/B and vancomycin, which were absent in the type strains. Another important pathway that was highlighted in these spaceflight isolates is the betalain biosynthesis pathway and the presence of gene 4,5-DOPA dioxygenase extradiol. The enzyme dioxygenase extradiol is involved in opening the cyclic ring of dihydroxy-phenylalanine (DOPA), resulting in the formation of an unstable seco-DOPA that further rearranges to form betalamic acid, the structural unit of betalains (51). Betalains are known to be majorly produced by *Caryophyllales* plants (e.g., the red and yellow pigments found in beets) and some fungal species; however, one bacterial species, *Gluconacetobacter diazotrophicus*, was recently reported to produce betalains (52). They are known to have antioxidant activity, are reported to be effective against some cancer cells, and also promote anti-inflammatory response (53–55).

In addition, KEGG Orthology analysis indicated the presence of some genes in *S. sanguinis* spaceflight isolates that may have been acquired to help them survive the use of antibiotics, as noted by multiple studies of other microbial species on the ISS (26, 56–59). For instance, the mercury resistance operon present in some bacteria contains mostly three genes: *merR* (transcriptional regulator), *merT* (inner membrane mercury scavenging protein), and *merP* (periplasmic mercury scavenging protein) (60), which were also found in *S. sanguinis* spaceflight and type strains. In addition to these three genes, the spaceflight-relevant isolates also showed the presence of *merB* (alkyl-mercury lyase), *merE* (mercuric ion transport protein), and *merD* (MerR family transcriptional regulator, mercuric resistance operon regulatory protein), which are not present in all bacteria containing the operon. Previous studies have reported that several *mer* genes (*merD* and *merE*) were more likely to be encoded on plasmids or transposons and, therefore, can be transferred via horizontal gene transfer (61), which might also be the case for *S. sanguinis* spaceflight isolates. *S. sanguinis* spaceflight isolates also contained gene *rcnA*, belonging to nickel/cobalt transporter family protein.

**Conclusions.** This study utilized comparative genomics and identified a plethora of factors in *Sphingomonas* present in spaceflight environments that may result in the production of several PGPFs and aid in the growth of plants. The production of one of the phytohormones IAA was also confirmed using metabolomics. These space *Sphingomonas* strains, along with other plant growth-promoting strains of microbes isolated from the ISS (62), therefore, hold great potential to contribute to space exploration through promoting and controlling plant growth as part of an integrated bio-regenerative life support system enabling future extended-duration crewed missions.

## MATERIALS AND METHODS

**Sample collection and whole-genome sequence analysis.** Sample collection and processing, followed by WGS and assembly that led to the isolation of *Sphingomonas* strains, are described elsewhere (25, 63). Samples collected across different locations on the ISS, and the KSC-PHSF cleanroom, were plated on Reasoner's 2A agar (R2A) to culture different bacterial colonies (25). In addition to cultivation, previously published metagenomic sequences sampled from ISS locations were downloaded from NCBI (26) and analyzed for the presence of *Sphingomonas* species, and MAGs were generated.

DNA extraction was followed by metagenomic sequencing and assembly as described previously (26). Identification of *Sphingomonas* spp. from metagenomic assembly was performed as described

elsewhere (59). In the case of *S. sanguinis* genome assembled from the metagenome, sample collection, processing, and DNA extraction were carried out as reported previously (26). The assembled draft genomes were annotated using NCBI Prokaryotic Genome Annotation Pipeline (PGAP) 4.11 (64, 65). The identity of the strains was confirmed based on ANI calculated using EZBioCloud (66).

**Genome annotation.** To further explore potential functional differences between spaceflight isolates and their respective type strains, we obtained coding sequences from the NCBI RefSeq database and, where applicable, performed additional annotation through the online eggNOG-mapperV2 web server using the default settings (67, 68). The coding sequences of the metagenome-assembled genome of *S. sanguinis* ISS-IIF7SWP were retrieved from the NCBI GenBank database. Coding nucleotide sequences for each *Sphingomonas* genome along with their respective type strains were annotated using the online eggNOG-MapperV2 software, whose column-based outputs were parsed and compared on a per-genome basis using a custom python script (available for use via https://github.com/jlombo96/Sphingomonas-EggNOG-Workbook). Preliminary annotations of these select spaceflight isolates highlighted potential connections to the plant endophyte *Sphingomonas* sp. LK11, a strain well characterized with respect to its plant growth-promoting capabilities (12, 15, 24, 27, 69). Given the presence of a thoroughly annotated and complete genome sequence (15), robust evidence for the production of the plant phytohormones gibberellic acid (GA) and IAA (12, 15), and previous use as a model for novel plant-associated *Sphingomonas* genomes (5), *Sphingomonas* sp. LK11 was selected as a reference genome to identify putative plant growth-promoting features in other *Sphingomonas* genomes. In this study, the endophyte *Sphingomonas* sp. LK11, two *S. sanguinis* isolates (IIF7SW-B3A and IIF7SW-B5), and one *S. sanguinis* MAG (ISS-IIF7SWP) sampled from the ISS, *S. paucimobilis* FKI-L5-BR-P1 cultured from KSC-PHSF, and *S. paucimobilis* LCT-SP1 isolated from the Chinese spacecraft Shenzhou X were compared against their respective type strains: *S. sanguinis* NBRC 13937$^T$ and *S. paucimobilis* NCTC 11030$^T$.

**Metabolite extraction and analysis.** Previous reports have characterized the phytohormone production capability of several *Sphingomonas* species, including *Sphingomonas* sp. LK11 (11), *S. panacis* DCY99$^T$ (5), and *S. paucimobilis* ZJSH1 (14). While both *S. paucimobilis* ZJSH1 and *Sphingomonas* sp. LK11 were confirmed to produce the plant phytohormone IAA, the definitive chemical pathway responsible for their production is yet to be elucidated (12, 14, 15). Therefore, the potential for the production of the auxin-derivative IAA in each of the selected *S. paucimobilis* and *S. sanguinis* strains was examined. The space isolates (*S. sanguinis* IIF7SW-B3A, *S. sanguinis* IIF7SW-B5, and *S. paucimobilis* FKI-L5-BR-P1) were streaked on R2A (Difco, USA) and incubated at 28°C for 5 days. The strains were then inoculated in nutrient broth (Sigma-Aldrich, USA) containing 3 g/L L-tryptophan (Sigma-Aldrich, USA) and incubated at 28°C, 180 rpm for 7 days. To examine the phytohormones produced and secreted in culture filtrate, we pelleted the cells and extracted the metabolites from the supernatant using ethyl acetate. This step was repeated twice. The culture supernatant was then acidified using 6N HCl and metabolites were extracted using ethyl acetate. The ethyl acetate layer after each step described above was transferred to a fresh tube and evaporated in TurboVap LV. The residue obtained after drying the solvent was resuspended in 5% dimethyl sulfoxide (DMSO)/95% (vol/vol) methanol at a final concentration of 1 mg/mL. A suitable aliquot of the extract (10 $\mu$L) was injected for HPLC-DAD-MS analysis.

Stock solution of IAA standard (Sigma-Aldrich, USA) was prepared in 5% DMSO/95% methanol (vol/vol) at a concentration of 1 mg/mL. It was diluted 10 times, and 10 $\mu$L of the dilution was injected for HPLC-DAD-MS analysis. HPLC-DAD-MS was carried out using Thermo Finnigan LCQ Advantage ion trap mass spectrometer with an RP C$_{18}$ column (Alltech Prevail C$_{18}$ 3 mm 2.1 by 100 mm) at a flow rate of 125 $\mu$L/min. The solvent gradient for liquid chromatography/mass spectrometry (LC/MS) was 95% (vol/vol) acetonitrile (MeCN)/H$_2$O (solvent B) in 5% MeCN/H$_2$O (vol/vol; solvent A), both containing 0.05% (vol/vol) formic acid, as follows: 0% solvent B from 0 to 5 min, 0% to 100% solvent B from 5 min to 30 min, 100% solvent B from 30 to 45 min, 100% to 0% solvent B from 45 to 45.10 min, and reequilibration with 0% solvent B from 45.10 to 50 min.

**Comparative functional analysis.** Genome similarities between spaceflight *Sphingomonas* strains, their respective type strains, and the plant endophyte *Sphingomonas* sp. LK11 were visualized using BRIG version 0.95 with the *Sphingomonas* sp. LK11 (chromosome and plasmids) assigned as the reference genome (70). Percent identity values for space isolates and corresponding type strains were calculated using blastn (E value: 10, lower identity cutoff: 70%, upper identity cutoff: 90%).

To address potential functional differences between space isolates and their corresponding type strains, custom Python and R scripts (https://github.com/jlombo96/Sphingomonas-EggNOG-Workbook) were designed to parse output annotation tables and summarize their features on a per-genome basis. Descriptions for select KEGG IDs identified in eggNOG-MapperV2 were assigned using the BioServices API (71). To identify protein-level differences between the *Sphingomonas* strains obtained from spaceflight-relevant surfaces and their respective type strains, custom python scripts were created to construct pileup gene count tables for the annotations generated from the EggNOG-MapperV2 software. Genes identified by eggNOG-MapperV2 for the *S. paucimobilis* and *S. sanguinis* spaceflight isolates were pooled on a species basis and then compared to their type strains using JVENN (72) and UpSetR (73) using intersection analyses. Output tables generated by these analyses are displayed in Table S1.

**Pangenome analysis.** Pangenome analysis of the *Sphingomonas* space isolates, their type strains, and the plant endophyte *Sphingomonas* sp. LK11 was performed using Anvi'o v6.2 (74–76). Primary functional annotation in the Anvi'o software was conducted using the anvi-script-FASTA-to-contigs-db and anvi-run-ncbi-cogs commands. Subsequent pangenome gene clustering was carried out using blastp via the anvi-pan-genome command (–num-threads 2, –mcl-inflation 6, –min-bit 0.5, –use-ncbi-blast). The ordering of the pangenome display was determined using Euclidean distances and ward linkage settings.

**Ethics approval and consent to participate.** Because no human subjects were characterized, ethics approval is not needed.

**Data availability.** All sequence data used in this study have been uploaded to NCBI GenBank under accession numbers JABEOW000000000, JABEOV000000000, JABEOU000000000, and JABYQV000000000.

## SUPPLEMENTAL MATERIAL

Supplemental material is available online only.

**SUPPLEMENTAL FILE 1**, XLSX file, 1 MB.
**SUPPLEMENTAL FILE 2**, PDF file, 0.3 MB.

## ACKNOWLEDGMENTS

Part of the research described in this publication was carried out at the Jet Propulsion Laboratory, California Institute of Technology, under a contract with NASA. We thank researchers associated with Biotechnology and Planetary Protection group at JPL for their facility support. Government sponsorship acknowledged.

This research was funded by 2012 Space Biology NNH12ZTT001N grant number 19-12829-26 under task order NNN13D111T awarded to K.V. The funders had no role in study design, data collection and interpretation, the writing of the manuscript, or the decision to submit the work for publication.

K.V., N.K.S., S.G., and R.B. formulated the concept; J.L. and S.B. executed the study. K.V. collected samples, isolated strains, and coordinated and designed the study with input from all authors. J.L. performed comparative genome analyses, interpreted the genomics data, and generated the corresponding figures and tables. S.B. performed metabolomic analysis and worked with N.K.S. in generating draft genome assembly. N.K.S. contributed to microbial identification, processed shotgun metagenome sequence data, and generated MAGs. J.L. and S.B. wrote the draft manuscript, and K.V., J.M.W., S.G., and R.B. edited the manuscript for content. All authors have read the manuscript, are responsible for data interpretation, and approved the manuscript.

Reference herein to any specific commercial product, process, or service by trade name, trademark, manufacturer, or otherwise does not constitute or imply its endorsement by the U.S. Government or the Jet Propulsion Laboratory, California Institute of Technology.

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
