## [Reviewer comments · Microbiology Spectrum]

Microbiology Spectrum

Genomic characterization of potential plant-growth promoting features of *Sphingomonas* strains isolated from the International Space Station

Jonathan Lombardino, Swati Bijlani, Nitin Singh, Jason Wood, Richard Barker, Simon Gilroy, Clay Wang, and Kasthuri Venkateswaran

Corresponding Author(s): Kasthuri Venkateswaran, California Institute of Technology

Review Timeline:

Submission Date:

November 18, 2021

Accepted:

December 1, 2021

Editor: Kim Handley

Reviewer(s): The reviewers have opted to remain anonymous.

Transaction Report:

DOI: <https://doi.org/10.1128/spectrum.01994-21>

December 1, 2021

Dr. Kasthuri Venkateswaran
California Institute of Technology
Jet Propulsion Laboratory
Mail Stop 245-105
4800, Oak Grove Dr.
Pasadena, CA 91101

Re: Spectrum01994-21 (Genomic characterization of potential plant-growth promoting features of Sphingomonas strains isolated from the International Space Station)

Dear Dr. Kasthuri Venkateswaran:

Your manuscript has been accepted, and I am forwarding it to the ASM Journals Department for publication. You will be notified when your proofs are ready to be viewed.

Sincerely,

Kim Handley
Editor, Microbiology Spectrum

Journals Department
Supplemental Fig S1: Accept
Supplemental Table S1: Accept